# Coherent Targets Parameter Estimation for EVS-MIMO Radar

**Xueke Ding [1], Ying Hu [2], Changming Liu [2] and Qun Wan [1,\*]**

1   School of Information and Communication Engineering, University of Electronic Science and Technology of China, Chengdu 611731, China
2   Tongfang Electronic Technology Co., Ltd., Jiujiang 332099, China
\*   Correspondence: wanqun@uestc.edu.cn; Tel.: +86-138-7925-8642

**Abstract:** As an emerging technique for detection, electromagnetic vector sensor multiple-input multiple-output (EVS-MIMO) radar has attracted extensive interest recently. This paper focuses on the coherent targets issue in EVS-MIMO radar, and a spatial smoothing estimator is developed to estimate the multiple parameters. It first recovers the rank of the array data via forward spatial smoothing. Then, it estimates the elevation angles via the rotational invariance technique. Combined with the vector cross-product method, the azimuth angles are obtained. Thereafter, with the previously achieved direction angles, the polarized parameters are acquired by using the least squares technique. Unlike the existing polarization smoothing techniques, the proposed estimator is able to estimate the two-dimensional direction parameter. Furthermore, it can provide a polarized parameter of the target. In addition, the proposed estimator is computationally efficient, since it offers closed-form and automatically paired solutions to all the parameters. Numerical experiments are carried out to show its superiority and effectiveness.

**Keywords:** EVS-MIMO radar; direction estimation; polarization; coherent target

## 1. Introduction

Multiple-input multiple-output (MIMO) technique, benefitting from spatial diversity, has gained huge success in wireless communications. In addition, MIMO technique has pointed out a promising future for radar detection [1–5]. A MIMO radar refers to a radar with multiple sensors for both the Tx (Tx) and receiving (Rx). Unlike a phased-array (PA) radar [6], each sensor of a MIMIO radar emits an individual waveform (and orthogonal to other Tx sensor). Owing to both spatial diversity and waveform diversity, a MIMO radar occupies more degree-of-freedom (DOF) than a PA one, thus offering much better detection accuracy than the latter.

As a critical task in a bistatic MIMO radar, to estimate the direction-of-departure (DOD) and direction-of-arrival (DOA) has been frequently discussed. To achieve super-resolution angle estimation, many efforts have been done and a lot of estimators have been proposed. Generally, the estimation algorithms for a MIMO radar are extended from sensor arrays, but the former are often more complex than the latter. Classical estimators for MIMO radar include estimation of parameters with rotational invariance technique (ESPRIT) [7,8], Capon, multiple signal classification (MUSIC) [9–11], propagator method (PM) [12], maximum-likelihood (ML) [13], and tensor method [14,15]. Generally speaking, the spectrum searching counterparts, e.g., MUSIC and ML, are inefficient. ESPRIT is economical from the perspective of calculation efficiency, because it offers closed-form solutions for parameter estimation. Benefitting from exploring the multi-dimension characteristic of the array measurement, the tensor-based algorithm always offers more accurate estimates than a matrix algorithm. The current algorithms mainly concentrate on the one-dimensional (1D) estimation problem, i.e., to estimate the 1D-DOD and 1D-DOA. In engineering applications, nevertheless, to estimate the two-dimensional (2D) direction is much more appealing. In order to pursue 2D angle estimation, non-linear Tx and Rx array

geometries (for instance, rectangular/circular array) must be utilized [16,17]. However, nonlinear frameworks are often much more complicated than the uniform linear array (ULA) architecture, and they may suffer from position errors.

Recently, electromagnetic vector sensor (EVS) has drawn extensive attention in target detection [18,19]. Compared with the scalar-sensor based MIMO radar, a simple ULA configured with EVS-MIMO is capable of providing 2D-DOD and 2D-DOA estimation. Besides, additional polarization information (Tx polarization angle, TPA, and Rx polarization angle, RPA) of the targets can be provided, which is important in identifying targets with the same DOD/DOA. The EVS-MIMO radar concept was firstly proposed in Ref. [20], in which a MIMO radar with Tx EVS array and a single Rx EVS is considered. The ULA-configured EVS-MIMO radar was introduced in Ref. [21], in which both the transmitters and receivers are EVS arrays with ULA geometries. Moreover, an ESPRIT-like approach was proposed to estimate the multi-dimensional parameters. Therein, the elevation angles were estimated via ESPRIT, while the azimuth angles were obtained via vector cross-product (VCP). To make full use of the effective aperture, an improved ESPRIT estimator was derived in Ref. [22]. Moreover, the PARAFAC estimator was presented in Ref. [23], which offers more accurate performance than those in Refs. [21–23]. A coprime array-based EVS-MIMO architecture was given in Ref. [24], which occupies a larger aperture than the ULA configuration. More recently, an arbitrary geometry-based EVS-MIMO architecture has been investigated in Ref. [25], and an ESPRIT estimator combined with normalized VCP method was carried out, which is insensitive to the Tx and Rx sensor's' localizations.

It should be emphasized that the algorithms in Refs. [20–25] are only suitable for scenarios with uncorrelated targets. In practice, correlated targets commonly appear and have been extensively discussed in MIMO radar. Up to now, however, only a few works have paid attention to de-correlation of the coherent targets in EVS-MIMO radar. In Ref. [26], the polarization smoothing (PS) method was proposed, which recovers the rank of the array data by averaging the array data in the polarization domain. Another PS estimator was presented in Ref. [27], in which a different smoothing idea is adopted to recover the rank of the noiseless data. Nevertheless, the algorithms in both Refs. [26,27] would sacrifice the polarization detection ability of an EVS-MIMO radar, and are hence unable to provide polarization state estimation of the targets. In Ref. [28], the generalized spatial smoothing (GSS) strategy was introduced, which is suitable for arbitrary array geometry. Nevertheless, since the smoothing requires the participation of all the Tx and Rx EVS, it suffers from the drawback of sacrificed visual aperture.

In this paper, a spatial smoothing algorithm is developed for a EVS-MIMO radar, which is able to deal with the coherent targets. Explicitly, the contributions of this paper are highlighted as follows:

(1) The EVS-MIMO radar is adopted to detect the coherent targets. Unlike the traditional scalar ULA-based MIMO radar, a ULA-configured EVS-MIMO radar can not only provide 2D-DOD and 2D-DOA estimation, but is also capable of offering 2D-TPA and 2D-RPA estimation. The former advantage enables an EVS-MIMO radar to provide three-dimensional positioning of the coherent targets, while the latter advantage may help the radar system to detect coherent targets with weak strength;

(2) A spatial smoothing approach is introduced to tackle the coherent targets in EVS-MIMO radar. It solves the rank-deficiency problem by the smoothing of the array measurements along the spatial direction. Unlike the GSS method in Ref. [28], only parts of the Tx and Rx EVS are needed in the smoothing procedure, hence it has less visual aperture loss. Consequently, the proposed algorithm should achieve more accurate estimation results than the GSS approach in Ref. [28];

(3) An ESPRIT-like idea is carried out for multiple parameter estimations from the smoothed array data. After performing eigendecomposition on the reduced co-variance matrix, the ESPRIT idea is adopted to estimate the elevation angles. Then the VCP method is adopted to obtain the Tx/Rx azimuth angles. After the 2D-DOD and 2D-DOA estimation has been accomplished, the 2D-TPA and 2D-RPA can be

easily obtained by using the least squares (LS) approach. The proposed algorithm offers closed-form results for angle and polarization parameters estimation, so it is computationally efficient;

(4) We provide theoretical analysis in terms of target identifiability and Cramér–Rao bound (CRB). In addition, the theoretical advantages of the proposed algorithms are verified via computer trials.

Throughout this paper, we use bold capital letters to denote matrix, and use bold lowercase letter to denote vector; The $M \times M$ identity matrix is denoted by $I_M$, while $0_{M \times N}$ and $1_M$ denote the $M \times N$ full zeros matrix and the $M \times M$ full ones matrix, respectively; $(X)^T$, $(X)^H$, $(X)^{-1}$, and $(X)^\dagger$, respectively, denote transpose, Hermitian transpose, inverse, and Pseudoinverse; $\odot$, $\otimes$ and $*$ denote the Khatri–Rao product, the Kronecker product, and the VCP, respectively; $\| \cdot \|$ denotes the Frobenius norm; the symbol $diag\{\cdot\}$ accounts for the diagonalization; $angle(\cdot)$ returns the phase; $E\{\cdot\}$ returns the mathematical expectation.

## 2. Problem Formulation

### 2.1. EVS Preliminaries

A full EVS is composed of six collocated components: mutual orthogonal magnetic loops (three) and electric dipoles (three), which sense the polarization state and the electronic field information, respectively. For a signal $s(t)$ impinging on an EVS, the response can be approximated by [18,19]

$$r(t) = bs(t), \tag{1}$$

where t is the snapshot index; $b \in \mathbb{C}^{6 \times 1}$ denotes the polarization steering vector, which is given by [18,19]

$$\begin{aligned} b &= \begin{bmatrix} e \\ p \end{bmatrix} \\ &= Dv \end{aligned} \tag{2}$$

with

$$D = \begin{bmatrix} \cos\phi\cos\theta & -\sin\phi \\ \sin\phi\cos\theta & \cos\phi \\ -\sin\theta & 0 \\ -\sin\phi & -\cos\phi\cos\theta \\ \cos\phi & -\sin\phi\cos\theta \\ 0 & \sin\theta \end{bmatrix} \tag{3}$$

and

$$v = \begin{bmatrix} \sin\gamma e^{j\eta} \\ \cos\gamma \end{bmatrix}, \tag{4}$$

where $e \in \mathbb{C}^{3 \times 1}$ and $p \in \mathbb{C}^{3 \times 1}$ account for the electric-field component and the polarization component, respectively; $\theta$, $\phi$, $\gamma$, and $\eta$ stand for, respectively, elevation, azimuth, auxiliary polarization, and polarized phase difference [18,19]. It should be emphasized that the normalized VCP between $e$ and $p$ fulfills

$$\frac{e}{|e|} * \frac{p^*}{|p|} = \begin{bmatrix} \sin\theta\cos\phi \\ \sin\theta\sin\phi \\ \cos\theta \end{bmatrix}. \tag{5}$$

For an $M$-element EVS array, the array response is given by

$$y(t) = [a \otimes b]s(t), \tag{6}$$

where $a \in \mathbb{C}^{M \times 1}$ denotes the spatial steering vector, which is associated with the array geometry.

*2.2. Data Model*

Consider a ULA-configured EVS-MIMO radar. Suppose that the EVS-MIMO radar is equipped with $M$ Tx EVS and $N$ Rx EVS, with both of the adjacent distances of them being $\lambda/2$, $\lambda$ denotes the wavelength of the carrier. If there are $K$ targets in the far-field of both of the Tx and Rx arrays, let the 2D-DOA, 2D-DOD, 2D-RPA, and 2D-TPA of the $k$-th target be $(\theta_{r,k}, \phi_{r,k})$, $(\theta_{t,k}, \phi_{t,k})$, $(\gamma_{r,k}, \eta_{r,k})$ and $(\gamma_{t,k}, \eta_{t,k})$, where $\theta_{t,k}$, $\theta_{r,k}$, $\phi_{t,k}$, and $\phi_{r,k}$ are the direction angles, as illustrated in Figure 1; $\gamma_{t,k}$, $\gamma_{r,k}$, $\eta_{t,k}$, and $\eta_{r,k}$ are the associated polarized parameters. The matched outputs are given by

$$\begin{aligned} \boldsymbol{x}(\tau) &= \sum_{k=1}^{K} [\boldsymbol{a}_{t,k} \otimes \boldsymbol{b}_{t,k} \otimes \boldsymbol{a}_{r,k} \otimes \boldsymbol{b}_{r,k}] s_k(\tau) + \boldsymbol{n}(\tau) \\ &= [\boldsymbol{A}_t \odot \boldsymbol{B}_t \odot \boldsymbol{A}_r \odot \boldsymbol{B}_r] \boldsymbol{s}(\tau) + \boldsymbol{n}(\tau) \end{aligned} \tag{7}$$

where $\tau$ accounts for the pulse index, $\boldsymbol{s}(\tau) = [s_1(\tau), s_2(\tau), \cdots, s_K(\tau)]^T$ represents the reflection coefficient vector of the targets; $\boldsymbol{a}_{t,k} = [1, e^{-j\pi\sin\theta_{t,k}}, \cdots, e^{-j(M-1)\pi\sin\theta_{t,k}}]^T$ denotes the $k$-th Tx spatial steering vector, $\boldsymbol{a}_{r,k} = [1, e^{-j\pi\sin\theta_{r,k}}, \cdots, e^{-j(N-1)\pi\sin\theta_{r,k}}]^T$ denotes the $k$-th Rx steering vector; $\boldsymbol{b}_{t,k} \in \mathbb{C}^{6\times K}$ denotes the $k$-th Tx polarization vector, and $\boldsymbol{b}_{r,k} \in \mathbb{C}^{6\times K}$ denotes the $k$-th Rx polarization vector; $\boldsymbol{A}_t = [\boldsymbol{a}_{t,1}, \boldsymbol{a}_{t,2}, \cdots, \boldsymbol{a}_{t,K}] \in \mathbb{C}^{M\times K}$, $\boldsymbol{A}_r = [\boldsymbol{a}_{r,1}, \boldsymbol{a}_{r,2}, \cdots, \boldsymbol{a}_{r,K}] \in \mathbb{C}^{N\times K}$; $\boldsymbol{B}_t = [\boldsymbol{b}_{t,1}, \boldsymbol{b}_{t,2}, \cdots, \boldsymbol{b}_{t,K}] \in \mathbb{C}^{6\times K}$, $\boldsymbol{B}_r = [\boldsymbol{b}_{r,1}, \boldsymbol{b}_{r,2}, \cdots, \boldsymbol{b}_{r,K}] \in \mathbb{C}^{6\times K}$; $\boldsymbol{n}(\tau)$ accounts for the zero-mean Gaussian white noise vector with a variance of $\sigma^2$. Besides,

$$\begin{cases} \boldsymbol{b}_{t,k} &= \begin{bmatrix} \boldsymbol{e}_{t,k} \\ \boldsymbol{p}_{t,k} \end{bmatrix} \\ &= \boldsymbol{D}_{t,k}\boldsymbol{v}_{t,k} \\ \boldsymbol{b}_{r,k} &= \begin{bmatrix} \boldsymbol{e}_{r,k} \\ \boldsymbol{p}_{r,k} \end{bmatrix} \\ &= \boldsymbol{D}_{r,k}\boldsymbol{v}_{r,k} \end{cases}, \tag{8}$$

where $\boldsymbol{D}_{t,k}$, $\boldsymbol{D}_{r,k}$, $\boldsymbol{v}_{t,k}$, and $\boldsymbol{v}_{r,k}$ are constructed according to (3) and (4), respectively. Define $\boldsymbol{C} = \boldsymbol{A}_t \odot \boldsymbol{B}_t \odot \boldsymbol{A}_r \odot \boldsymbol{B}_r$, and the covariance of $\boldsymbol{x}(\tau)$ in (7) is then formulated as

$$\begin{aligned} \boldsymbol{R} &= E\{\boldsymbol{x}(\tau)\boldsymbol{x}^H(\tau)\} \\ &= \boldsymbol{C}\boldsymbol{R}_s\boldsymbol{C}^H + \sigma^2 \boldsymbol{I}_{36MN} \end{aligned} \tag{9}$$

where $\boldsymbol{R}_s = E\{\boldsymbol{s}(\tau)\boldsymbol{s}^H(\tau)\}$ is the covariance matrix of the target reflection coefficients. When the targets are uncorrelated, $\boldsymbol{R}_s$ is diagonal. Since the identity matrix would not affect the orthogonal decomposition of a matrix, the signal subspace can maintain the relationship between the multiple components of $C$. Thus, the eigendecomposition is firstly performed on $R$ to obtain the signal subspace. However, in the presence of correlated targets, $\boldsymbol{R}_s$ is rank deficient. Therefore, the subspace method will fail to work.

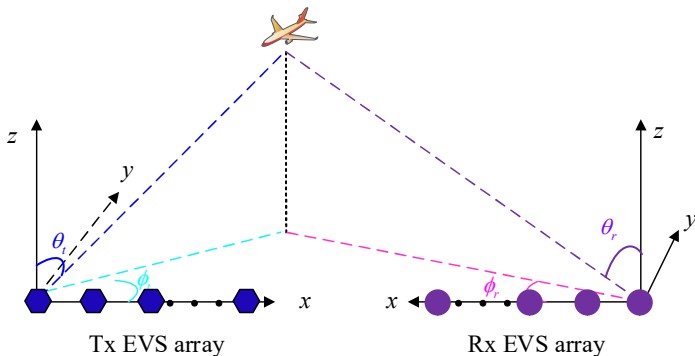

**Figure 1.** Schematic diagram of a bistatic EMVS-MIMO radar with ULA-configured Tx/Rx array.

### 3. The Proposed Approach

*3.1. Spatial Smoothing for EVS-MIMO Radar*

In this subsection, we will show how the spatial smoothing is applied to the EVS-MIMO radar. For the Tx array, suppose that a subarray consists of the first $M_1$-element EVS ($1 < M_1 < M$), so the matched filtering results corresponding to the Tx subarray and the first Rx EVS can be formulated as

$$\begin{aligned} x_{t1,r0}(\tau) &= \sum_{k=1}^{K} [a_{t1,k} \otimes b_{t,k} \otimes b_{r,k}]s_k(\tau) + n_{t1,r0}(\tau) \\ &= [A_{t1} \odot B_t \odot B_r]s(\tau) + n_{t1,r0}(\tau) \end{aligned} \tag{10}$$

where $a_{t1,k}$ and $A_{t1}$ are the steering vector and direction matrix corresponding to the Tx subarray. $n_{t1,r0}(\tau)$ denotes the associated array noise. The covariance matrix of $x_{t1,r0}(\tau)$ is

$$\begin{aligned} R_{t1,r0} &= E\left\{x_{t1,r0}(\tau)x_{t1,r0}^H(\tau)\right\} \\ &= C_{t1,r0}R_s C_{t1,r0}^H + \sigma^2 I_{36M_1} \end{aligned} \tag{11}$$

where $C_{t1,r0} = A_{t1} \odot B_t \odot B_r$. By moving the Tx EVS array forward, we can construct another $M - M_1 + 1$ Tx subarray. Likewise, we can pick up the array measurement corresponding to the m-th ($m = 1, 2, \cdots, M - M_1 + 1$) Tx EVS array and the first Rx EVS as

$$\begin{aligned} x_{tm,r0}(\tau) &= \sum_{k=1}^{K} [a_{tm,k} \otimes b_{t,k} \otimes b_{r,k}]s_k(\tau) + n_{t1,r0}(\tau) \\ &= [A_{tm} \odot B_t \odot B_r]s(\tau) + n_{t1,r0}(\tau) \end{aligned} \tag{12}$$

In fact, the relationship between $A_{tm}$ and $A_{t1}$ can be formulated as

$$A_{tm} = A_{t1}\Gamma_t^{-m}, \tag{13}$$

where $\Gamma_t = diag\left\{e^{j\pi \sin \theta_{t,1}}, e^{j\pi \sin \theta_{t,2}}, \cdots, e^{j\pi \sin \theta_{t,K}}\right\}$. Consequently, the covariance matrix of $x_{tm,r0}(\tau)$ is

$$\begin{aligned} R_{tm,r0} &= E\left\{x_{tm,r0}(\tau)x_{tm,r0}^H(\tau)\right\} \\ &= C_{t1,r0}\Gamma_t^{-m}R_s\left(\Gamma_t^{-m}\right)^H C_{t1,r0}^H + \sigma^2 I_{36M_1} \end{aligned} \tag{14}$$

By averaging all the $R_{tm,r0}$, we can obtain

$$\begin{aligned} R_{t,r0} &= \frac{1}{M-M_1+1} \sum_{m=1}^{M-M_1+1} R_{tm,r0} \\ &= C_{t1,r0}R_{s,t}C_{t1,r0}^H + \sigma^2 I_{36M_1} \end{aligned} \tag{15}$$

where

$$R_{s,t} = \frac{1}{M-M_1+1} \sum_{m=1}^{M-M_1+1} \Gamma_t^{-m}R_s\left(\Gamma_t^{-m}\right)^H. \tag{16}$$

It has been proven that if the angles are distinct, then $R_{s,t}$ is full rank, which implies that the rank of the noiseless $R_{t,r0}$ is K. However, it should be pointed out that for a rank-one matrix $R_s$, after the above forward smoothing, the maximum rank of $R_{s,t}$ is $M - M_1 + 1$, which implies that the above smoothing can resolve $M - M_1 + 1$ coherent targets at most. To enhance the ability for a coherent target, we can extend the forward smoothing by considering both of the Tx and Rx arrays. We consider that a Rx subarray consists of the first $N_1$-element EVS. Likewise, we can construct $N - N_1 + 1$ Rx subarray. Let $x_{t1,r1}(\tau)$

account for the matched array data associated with the first Tx subarray and the first Rx subarray, i.e.,

$$
\begin{aligned}
\boldsymbol{x}_{t1,r1}(\tau) &= \sum_{k=1}^{K} [\boldsymbol{a}_{t1,k} \otimes \boldsymbol{b}_{t,k} \otimes \boldsymbol{a}_{r1,k} \otimes \boldsymbol{b}_{r,k}] s_k(\tau) + \boldsymbol{n}_{t1,r1}(\tau) \\
&= [\boldsymbol{A}_{t1} \odot \boldsymbol{B}_t \odot \boldsymbol{A}_{r1} \odot \boldsymbol{B}_r] \boldsymbol{s}(\tau) + \boldsymbol{n}_{t1,r1}(\tau)
\end{aligned}
\tag{17}
$$

where $\boldsymbol{a}_{r1,k}$ and $\boldsymbol{A}_{r1}$ are the steering vector and direction matrix corresponding to the first Rx subarray, and $\boldsymbol{n}_{t1,r1}(\tau)$ is the associated array noise. The covariance matrix of $\boldsymbol{x}_{t1,r1}(\tau)$ is then given by

$$
\begin{aligned}
\boldsymbol{R}_{t1,r1} &= E\left\{\boldsymbol{x}_{t1,r1}(\tau)\boldsymbol{x}_{t1,r1}^{H}(\tau)\right\} \\
&= \boldsymbol{C}_{t1,r1}\boldsymbol{R}_s\boldsymbol{C}_{t1,r1}^{H} + \sigma^2 \boldsymbol{I}_{36M_1N_1}
\end{aligned}
\tag{18}
$$

where $\boldsymbol{C}_{t1,r1} = \boldsymbol{A}_{t1} \odot \boldsymbol{B}_t \odot \boldsymbol{A}_{r1} \odot \boldsymbol{B}_r$. Similarly, the array data corresponding to the $m$-th Tx EVS array and the $n$-th ($n = 1, 2, \cdots, N - N_1 + 1$) Rx EVS array is given by

$$
\begin{aligned}
\boldsymbol{x}_{tm,rn}(\tau) &= \sum_{k=1}^{K} [\boldsymbol{a}_{tm,k} \otimes \boldsymbol{b}_{t,k} \otimes \boldsymbol{a}_{rn,k} \otimes \boldsymbol{b}_{r,k}] s_k(\tau) + \boldsymbol{n}_{tm,rn}(\tau) \\
&= [\boldsymbol{A}_{t1} \odot \boldsymbol{B}_t \odot \boldsymbol{A}_{r1} \odot \boldsymbol{B}_r] \boldsymbol{\Gamma}_t^{-m}\boldsymbol{\Gamma}_r^{-n}\boldsymbol{s}(\tau) + \boldsymbol{n}_{tm,rn}(\tau)
\end{aligned}
\tag{19}
$$

where $\boldsymbol{\Gamma}_r = diag\{e^{j\pi \sin\theta_{r,1}}, e^{j\pi \sin\theta_{r,2}}, \cdots, e^{j\pi \sin\theta_{r,K}}\}$, $\boldsymbol{a}_{rn,k}$ is the associated steering vector, and $\boldsymbol{n}_{tm,rn}(\tau)$ is the corresponding array noise. Consequently, the covariance of $\boldsymbol{x}_{tm,rn}(\tau)$ is formulated as

$$
\begin{aligned}
\boldsymbol{R}_{tm,rn} &= E\{\boldsymbol{x}_{tm,rn}(\tau)\boldsymbol{x}_{tm,rn}^{H}(\tau)\} \\
&= \boldsymbol{C}_{t1,r1}\boldsymbol{\Gamma}_t^{-m}\boldsymbol{\Gamma}_r^{-n}\boldsymbol{R}_s\left(\boldsymbol{\Gamma}_t^{-m}\boldsymbol{\Gamma}_r^{-n}\right)^{H}\boldsymbol{C}_{t1,r1}^{H} + \sigma^2 \boldsymbol{I}_{36M_1N_1}
\end{aligned}
\tag{20}
$$

By averaging all the $\boldsymbol{R}_{tm,rn}$, a covariance matrix $\boldsymbol{R}_{t,r}$ is constructed as

$$
\begin{aligned}
\boldsymbol{R}_{t,r} &= \frac{1}{M-M_1+1}\frac{1}{N-N_1+1}\sum_{m=1}^{M-M_1+1}\sum_{n=1}^{N-N_1+1}\boldsymbol{R}_{tm,rn} \\
&= \boldsymbol{C}_{t1,r1}\tilde{\boldsymbol{R}}_s\boldsymbol{C}_{t1,r0}^{H} + \sigma^2 \boldsymbol{I}_{36M_1N_1}
\end{aligned}
\tag{21}
$$

where

$$
\tilde{\boldsymbol{R}}_s = \frac{1}{M-M_1+1}\frac{1}{N-N_1+1}\sum_{m=1}^{M-M_1+1}\sum_{n=1}^{N-N_1+1}\boldsymbol{\Gamma}_t^{-m}\boldsymbol{\Gamma}_r^{-n}\boldsymbol{R}_s\left(\boldsymbol{\Gamma}_t^{-m}\boldsymbol{\Gamma}_r^{-n}\right)^{H}.
\tag{22}
$$

Similar to our previous conclusion, for a rank-one matrix $\boldsymbol{R}_s$, the maximum rank of $\tilde{\boldsymbol{R}}_s$ is $(M-M_1+1)(N-N_1+1)$, which implies that the above smoothing approach can identify $(M-M_1+1)(N-N_1+1)$ complete coherent targets. In fact, the estimates of $\boldsymbol{R}_{t,r}$ can be obtained from $L$ samples as

$$
\hat{\boldsymbol{R}}_{t,r} = \frac{1}{L}\frac{1}{M-M_1+1}\frac{1}{N-N_1+1}\sum_{m=1}^{M-M_1+1}\sum_{n=1}^{N-N_1+1}\sum_{t=1}^{L}\boldsymbol{x}_{tm,rn}(\tau)\boldsymbol{x}_{tm,rn}^{H}(\tau).
\tag{23}
$$

From the eigendecomposition of $\hat{\boldsymbol{R}}_{t,r}$ we could obtain the so-called signal subspace $\boldsymbol{E}_s$, which are picked up from the eigen-vectors associated with the $K$ largest eigen-values. As is well known to us, $\boldsymbol{E}_s$ and $\boldsymbol{C}_{t1,r1}$ span the same subspaces. As a result, there must be a matrix $\boldsymbol{T} \in \mathbb{C}^{K \times K}$ with full-rank, which fulfills

$$
\boldsymbol{E}_s = \boldsymbol{C}_{t1,r1}\boldsymbol{T}.
\tag{24}
$$

*3.2. 2D-DOD and 2D-DOA Estimation*

Define $J_1 = \begin{bmatrix} I_{M_1-1}, 0_{(M_1-1)\times 1} \end{bmatrix} \in \mathbb{C}^{(M_1-1)\times M_1}$, $J_2 = \begin{bmatrix} 0_{(M_1-1)\times 1}, I_{M_1-1} \end{bmatrix} \in \mathbb{C}^{(M_1-1)\times M_1}$, which select the first $M_1 - 1$ row and the last $M_1 - 1$ row of $A_{t1}$, respectively. Similarly, define $J_3 = \begin{bmatrix} I_{N_1-1}, 0_{(N_1-1)\times 1} \end{bmatrix} \in \mathbb{C}^{(N_1-1)\times N}$, $J_4 = \begin{bmatrix} 0_{(N_1-1)\times 1}, I_{N_1-1} \end{bmatrix} \in \mathbb{C}^{(N_1-1)\times N}$, which select the first $N_1 - 1$ row and the last $N_1 - 1$ row of $A_{r1}$, respectively. It is easy to find that

$$\begin{cases} J_1 A_{t1} = J_2 A_{t1} \Gamma_t \\ J_3 A_{r1} = J_4 A_{r1} \Gamma_r \end{cases}. \tag{25}$$

Next, define $J_{t1} = \begin{bmatrix} J_1 \otimes I_{36N_1} \end{bmatrix} \in \mathbb{C}^{36(M_1-1)N_1 \times 36M_1 N_1}$, $J_{t2} = \begin{bmatrix} J_2 \otimes I_{36N_1} \end{bmatrix} \in \mathbb{C}^{36(M_1-1)N_1 \times 36M_1 N_1}$, $J_{r1} = \begin{bmatrix} I_{6M_1} \otimes J_3 \otimes I_6 \end{bmatrix} \in \mathbb{C}^{36M_1(N_1-1)\times 36M_1 N_1}$, $J_{r2} = \begin{bmatrix} I_{6M_1} \otimes J_4 \otimes I_6 \end{bmatrix} \in \mathbb{C}^{36M_1(N_1-1)\times 36M_1 N_1}$. The rotational invariant property in (25) can be extended to

$$\begin{cases} J_{t1} C_{t1,r1} = J_{t2} C_{t1,r1} \Gamma_t \\ J_{r1} C_{t1,r1} = J_{r2} C_{t1,r1} \Gamma_r \end{cases}. \tag{26}$$

Inserting (26) into (24) yields

$$\begin{cases} J_{t1} E_s = J_{t2} E_s T^{-1} \Gamma_t T \\ J_{r1} E_s = J_{r2} E_s T^{-1} \Gamma_r T \end{cases}. \tag{27}$$

Equivalently,

$$\begin{cases} (J_{t2} E_s)^\dagger J_{t1} E_s = T^{-1} \Gamma_t T \\ T (J_{r2} E_s)^\dagger J_{r1} E_s T^{-1} = \Gamma_r \end{cases}. \tag{28}$$

One can easily find that the eigendecomposition of $(J_{t2} E_s)^\dagger J_{t1} E_s$ yields the estimations of $T$ and $\Gamma_t$ (denoted by $\hat{T}$ and $\hat{\Gamma}_t$, respectively). After that, we can achieve the estimations of $\Gamma_r$ (denoted by $\hat{\Gamma}_r$) by calculating the left side of the second item in (28). Denote the $k$-th diagonal elements with respect to $\hat{\Gamma}_t$ and $\hat{\Gamma}_r$ by $\hat{\lambda}_t$ and $\hat{\lambda}_r$, respectively. Then the estimates of $\theta_{t,k}$ and $\theta_{r,k}$ are given by

$$\begin{cases} \hat{\theta}_{t,k} = \arcsin\{angle(\hat{\lambda}_{t,k})/\pi\} \\ \hat{\theta}_{r,k} = \arcsin\{angle(\hat{\lambda}_{r,k})/\pi\} \end{cases}. \tag{29}$$

In what follows, we will concentrate on $\phi_{t,k}$ and $\phi_{r,k}$. From the relation in (24), we can estimate $C_{t1,r1}$ via

$$\hat{C}_{t1,r1} = E_s \hat{T}^{-1}. \tag{30}$$

Since the azimuth angles are related to the polarization steering vectors $b_{t,k}$ and $b_{r,k}$, we need to estimate them from $\hat{C}_{t1,r1}$. Define $J_5 = \begin{bmatrix} i_{M_1,p} \otimes I_6 \otimes i_{N_1,q} \end{bmatrix} \in \mathbb{C}^{6 \times 36M_1 N_1}$, $J_6 = \begin{bmatrix} i_{6M_1 N_1,p} \otimes I_6 \end{bmatrix} \in \mathbb{C}^{6 \times 36M_1 N_1}$, where $p$ and $q$ are arbitrary integers. Then we have

$$\begin{cases} J_5 C_{t1,r1} = B_t \Theta_t \\ J_6 C_{t1,r1} = B_r \Theta_r \end{cases}, \tag{31}$$

where $\Theta_t = \mathcal{F}_p(A_{t1})\mathcal{F}_q(A_{r1} \otimes B_r)$, $\Theta_r = \mathcal{F}_p(A_{t1} \otimes B_t \otimes A_{r1})$, $\mathcal{F}_p(A_{r1})$ denotes a diagonal matrix whose nonzero elements comes from the $p$-th row of $A_{r1}$, and is similar to others. Then we can estimate $B_t$ and $B_r$ via

$$\begin{cases} \hat{B}_t = J_5 \hat{C}_{t1,r1} \\ \hat{B}_r = J_6 \hat{C}_{t1,r1} \end{cases}. \tag{32}$$

Obviously, $\hat{B}_t$ and $\hat{B}_r$ are the estimations of $B_t\Theta_t$ and $B_r\Theta_r$, respectively. Let the $k$-th column of $\hat{B}_t$ and $\hat{B}_r$ be $\hat{b}_{t,k}$ and $\hat{b}_{r,k}$, and denote the first and last three entities of them by $\hat{e}_{t,k}$, $\hat{p}_{t,k}$, $\hat{e}_{r,k}$ and $\hat{p}_{r,k}$, respectively. According to (5), we have

$$
\begin{bmatrix} u_{t,k} \\ v_{t,k} \\ w_{t,k} \end{bmatrix} = \begin{bmatrix} \sin\hat{\theta}_{t,k}\cos\hat{\phi}_{t,k} \\ \sin\hat{\theta}_{t,k}\sin\hat{\phi}_{t,k} \\ \cos\hat{\theta}_{t,k} \end{bmatrix}
= \frac{\hat{e}_{t,k}}{|\hat{e}_{t,k}|} * \frac{\hat{p}_{t,k}}{|\hat{p}_{t,k}|}
\tag{33}
$$

and

$$
\begin{bmatrix} u_{r,k} \\ v_{r,k} \\ w_{r,k} \end{bmatrix} = \begin{bmatrix} \sin\hat{\theta}_{r,k}\cos\hat{\phi}_{r,k} \\ \sin\hat{\theta}_{r,k}\sin\hat{\phi}_{r,k} \\ \cos\hat{\theta}_{r,k} \end{bmatrix}
= \frac{\hat{e}_{r,k}}{|\hat{e}_{r,k}|} * \frac{\hat{p}_{r,k}^{*}}{|\hat{p}_{r,k}|}
\tag{34}
$$

Consequently, we can obtain the azimuth angles via

$$
\begin{cases} \hat{\phi}_{t,k} = \arctan(v_{t,k}/u_{t,k}) \\ \hat{\phi}_{r,k} = \arctan(v_{r,k}/u_{r,k}) \end{cases}.
\tag{35}
$$

Notably, as the non-singular matrix $T$ has been compensated, $\hat{\theta}_{t,k}$, $\hat{\theta}_{r,k}$, $\hat{\phi}_{t,k}$, and $\hat{\phi}_{r,k}$ are automatically paired.

### 3.3. 2D-TPA and 2D-RPA Estimation

After we have obtained $\hat{\theta}_{t,k}$, $\hat{\theta}_{r,k}$, $\hat{\phi}_{t,k}$, the matrices $\hat{\mathbf{D}}_{t,k}$ and $\hat{\mathbf{D}}_{t,k}$ can be constructed by referencing to (3). Thereafter, we calculate

$$
\begin{cases} \hat{\mathbf{v}}_{t,k} = \hat{\mathbf{D}}_k^{\dagger}\hat{\mathbf{b}}_k \\ \hat{\mathbf{v}}_{r,k} = \hat{\mathbf{D}}_k^{\dagger}\hat{\mathbf{b}}_k \end{cases}.
\tag{36}
$$

From what we have pointed out previously, $\hat{\mathbf{v}}_{t,k}$ and $\hat{\mathbf{v}}_{r,k}$ are the estimates of $z_{t,k}\mathbf{v}_{t,k}$ and $z_{r,k}\mathbf{v}_{r,k}$, respectively, where $z_{t,k}$ and $z_{r,k}$ are non-zero constants. Consequently, the estimates of $\gamma_{t,k}$, $\gamma_{r,k}$, $\eta_{t,k}$, and $\eta_{r,k}$, respectively, can be obtained via

$$
\begin{cases} \hat{\gamma}_{t,k} = \arctan\left(\hat{\mathbf{v}}_{t,k}(2)/\hat{\mathbf{v}}_{t,k}(1)\right) \\ \hat{\eta}_{t,k} = \mathrm{angle}\left(\hat{\mathbf{v}}_{t,k}(2)/\hat{\mathbf{v}}_{t,k}(1)\right) \\ \hat{\gamma}_{r,k} = \arctan\left(\hat{\mathbf{v}}_{r,k}(2)/\hat{\mathbf{v}}_{r,k}(1)\right) \\ \hat{\eta}_{r,k} = \mathrm{angle}\left(\hat{\mathbf{v}}_{r,k}(2)/\hat{\mathbf{v}}_{r,k}(1)\right) \end{cases}.
\tag{37}
$$

As explained previously, all the perturbations are synchronously completed, so the estimated 2D-TPA and 2D-RPA are automatically paired with the 2D-DOD as well as 2D-DOA.

## 4. Algorithm Analyses

### 4.1. Identifiability

It has been stressed that our algorithm is able to deal with a coherent target, thus the identifiability of the proposed algorithm should be analyzed from two aspects: the identifiability of the coherent target and the identifiability of the non-coherent target. For the coherent target, the maximum identifiable number equals to the maxi-

mum rank of $\widetilde{\boldsymbol{R}}_s$ for the rank-one $\boldsymbol{R}_s$, which is $(M - M_1 + 1)(N - N_1 + 1)$ (suppose that $(M - M_1 + 1)(N - N_1 + 1) \leq K$). On the other hand, the maximum detectable target number of the non-coherent target is constrained by the rank of $\boldsymbol{\Theta}_t$ (or $\boldsymbol{\Theta}_r$), which is $\min\{36(M_1 - 1)N_1, 36M_1(N_1 - 1)\}$. In contrast, the PS algorithm in Ref. [26] can detect at most $MN$ non-coherent targets and 72 coherent targets, while the abilities with respect to the algorithm in Refs. [26–28] are $MN$, 36, and 36, $\min\{MN, 36\}$, respectively. Table 1 lists the identifiability comparison of the various approaches. Obviously, the proposed algorithm has better identifiability than Refs. [26,27].

**Table 1.** Comparison of the identifiability.

| Method | Identifiability | |
|---|---|---|
| | **Non-Coherent Target** | **Coherent Target** |
| Algorithm in Ref. [26] | $MN$ | 72 |
| Algorithm in Ref. [27] | $MN$ | 36 |
| Algorithm in Ref. [28] | 36 | $\min\{MN, 36\}$ |
| Proposed | $\min\{36(M_1 - 1)N_1, 36M_1(N_1 - 1)\}$ | $(M - M_1 + 1)(N - N_1 + 1)$ |

From the above comparison, one can see that our algorithm has much better identifiability than that in Refs. [26,27], especially in the presence of massive MIMO configurations. Besides, it should be emphasized that both Refs. [26,27] cannot offer 2D direction angle, while the proposed algorithm can not only provide 2D elevation/azimuth angle estimation, but also 2D polarized parameters estimation, which is more flexible than Refs. [26,27].

*4.2. CRB*

The CRB on 2D-DOD, 2D-DOA, 2D-TPA, and 2D-RPA estimation is

$$CRB = \frac{\sigma^2}{2L}\left[real\left(\left(\widetilde{\boldsymbol{C}}^H \boldsymbol{\Pi}_{\boldsymbol{C}}^{\perp} \widetilde{\boldsymbol{C}}\right) \oplus \left(\boldsymbol{R}_s^T \otimes 1_{8 \times 8}\right)\right)\right]^{-1}, \tag{38}$$

with $\widetilde{\boldsymbol{C}} = \left[\frac{\partial c_1}{\partial \phi_{t,1}}, \cdots, \frac{\partial c_K}{\partial \phi_{t,K}}, \frac{\partial c_1}{\partial \theta_{t,1}}, \cdots, \frac{\partial c_K}{\partial \gamma_{r,K}}\right]$, where $\boldsymbol{c}_k$ is the k-th column of $\boldsymbol{C}$, $\boldsymbol{\Pi}_{\boldsymbol{C}}^{\perp} = \boldsymbol{I}_{36MN} - \boldsymbol{C}\left(\boldsymbol{C}^H \boldsymbol{C}\right)\boldsymbol{C}^H$, $1_{8 \times 8}$ denotes a $8 \times 8$ full ones matrix.

**5. Simulation Results**

Herein, we use the Monte Carlo method to assess the estimation performance. We consider that an EVS-MIMO radar is configured with $M$ EVS transmitters $N$ EVS receivers, and the Tx/Rx array is distributed in half-wavelength spaced ULA geometry. The data model utilized to the simulation is given in (7). Unless otherwise specified, assume there are $K= 4$ coherent targets, with parameter pairs, respectively, $\theta_t = (10°, 36°, 55°, 78°)$, $\phi_t = (15°, 50°, 30°, -53°)$, $\gamma_t = (10°, 52°, 22°, 75°)$, $\eta_t = (18°, -31°, 0°, 67°)$, $\theta_r = (5°, 45°, 20°, 78°)$, $\phi_r = (15°, -23°, -67°, 30°)$, $\gamma_r = (32°, 10°, 51°, 65°)$ and $\eta_r = (16°, 56°, 38°, 78°)$. We assume that the number of Tx EMVS and Rx EMVS of the two subarrays are $M_1$ and $N_1$, respectively. Moreover, suppose that there are $L$ samples. Each figure relies on 200 independent experiments. The signal-to-noise ratio (SNR) and root mean square error (RMSE) are defined as the same to that in Ref. [28].

**Example 1** . *We show the scattering results of our algorithm with M = 6, N = 8, $M_1$ = 3, $N_1$ = 4, and L = 500. Figure 2 shows the result with SNR = 0 dB. Obviously, the proposed algorithm can accurately estimate all the parameters and correctly pair them.*

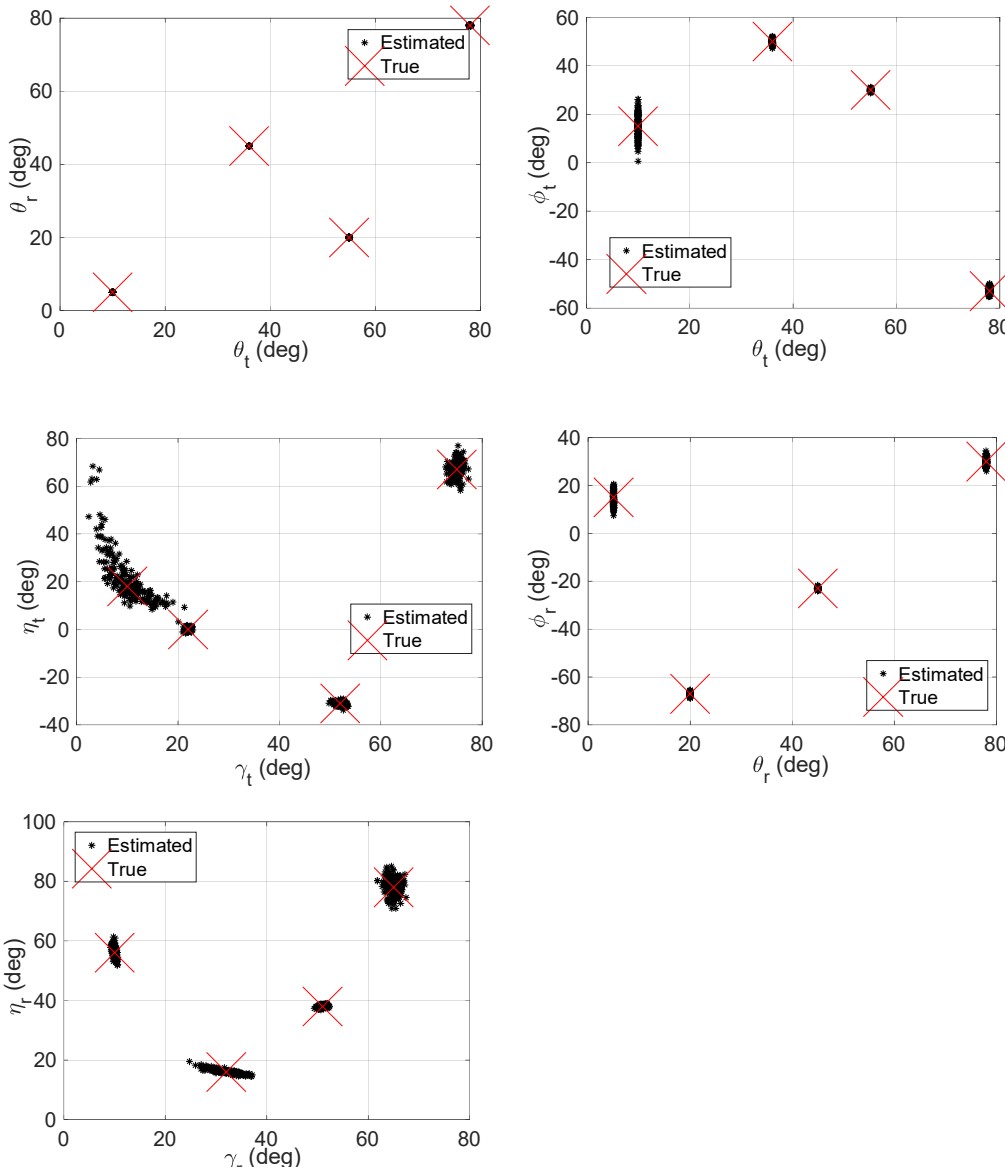

**Figure 2.** Scatters of our approach with $M = 6$, $N = 8$, $M_1 = 3$, $N_1 = 4$, $L = 500$, and SNR = 0 dB.

**Example 2** . *We repeat the scattering simulation with SNR = 20 dB, the other parameters are set to the same to that in Example 1. Figure 3 indicates the results. Notably, the proposed algorithm provides closed-form, automatically paired parameter estimates. Besides, it seems that higher SNR will lead to more concentrated scatter results.*

**Example 3** . *We evaluate the estimation accuracy of our algorithm with different SNR, where $M = 6$, $N = 8$, $M_1 = 3$, $N_1 = 4$, and $L = 500$. For the purpose of comparison, the performances with respect to the PARAFAC estimator in Ref. [23] (which has been proven to achieve the best estimation accuracy in the presence of uncoherent targets), the GSS approach in Ref. [28], and the CRB have been used. The average RMSE performances on 2D-DOD and 2D-DOA estimation are shown in Figure 4. It is indicated that the estimation accuracy of GSS and the proposed algorithm will be improved with the increasing SNR while RMSE of PARAFAC barely changes with SNR. Besides, the accuracy of the proposed estimator is much better than the GSS approach at low SNR regions, e.g., SNR is smaller than 10 dB, and the proposed algorithm is slightly better than GSS at high SNR regions. The improvement benefits from the truth that the proposed algorithm has larger visual aperture than the GSS approach, as we have previously stated.*

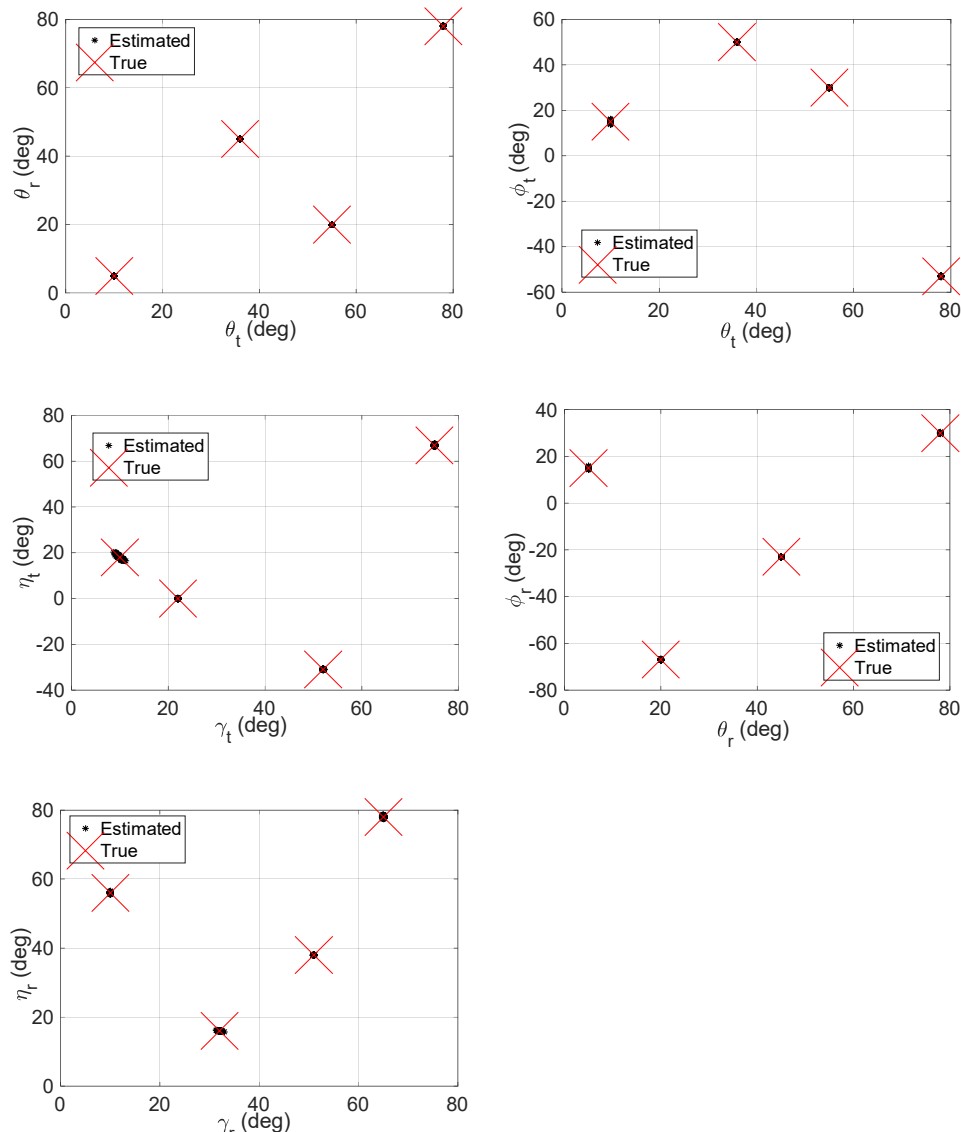

**Figure 3.** Scatters of our approach with *M* = 6, *N* = 8, $M_1 = 3$, $N_1 = 4$ , *L* = 500, and SNR = 20 dB.

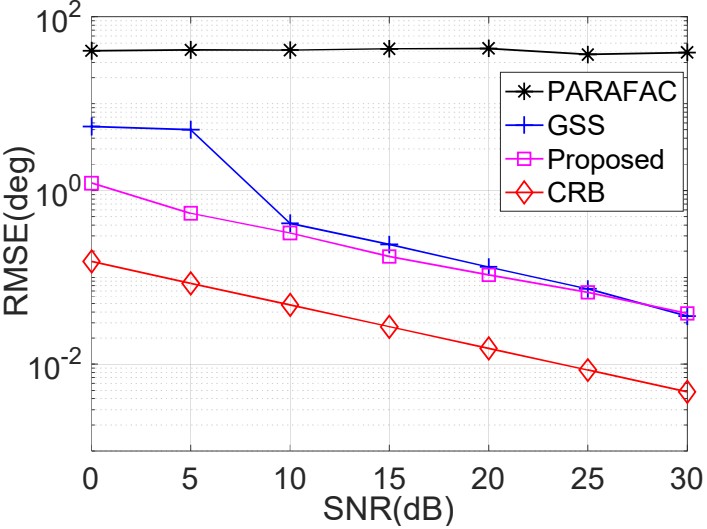

**Figure 4.** RMSE performance versus SNR.

**Example 4** . *We give the RMSE comparison versus Rx EVS number N, where M = 6, $M_1$ = 3, $N_1$ = 4, L = 500 and SNR = 15 dB. The result is shown in Figure 5. It is seen that the PARAFAC estimator fails to work for a coherent target. The proposed algorithm offers lower RMSE than the GSS approach when N is larger than 8, which implies that the proposed estimator is superior to GSS for massive Rx configuration.*

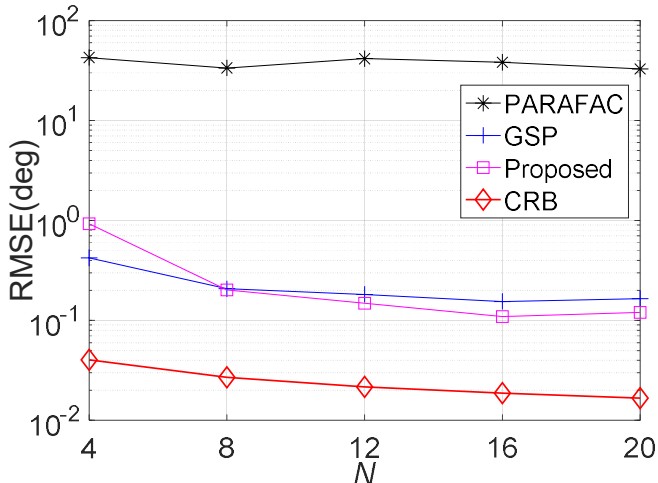

**Figure 5.** RMSE performance versus *N*.

**Example 5** . *The average RMSE comparison versus snapshot number L is displayed in Figure 6, where M = 6, N = 8, $M_1$ = 3 , $N_1$ = 4 and SNR = 15 dB. We can see that larger L will improve the estimation accuracy, except for the PARAFAC estimator. Obviously, the proposed algorithm outperforms the GSS approaches during the entire L regions, especially in the presence of small L values, e.g., when L is smaller than 80.*

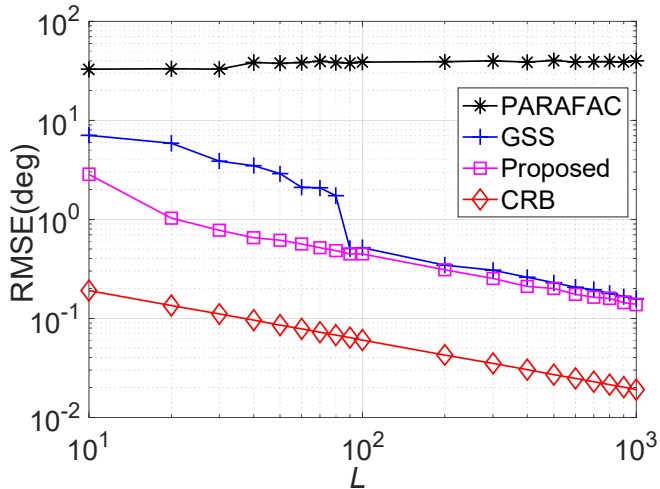

**Figure 6.** RMSE performance versus *L*.

**Example 6** . *The average RMSE comparison versus coherent target number K is illustrated in Figure 7, where M = 6, N = 8, $M_1$ = 3, $N_1$ = 4, L = 500, and SNR = 15 dB. Therein, K coherent target parameters are randomly generated. One can observed that a larger target number leads to less accurate estimation results. As expected, the proposed algorithm provides better estimation accuracy than GSS when K is larger than 5, since the former occupies larger visual aperture.*

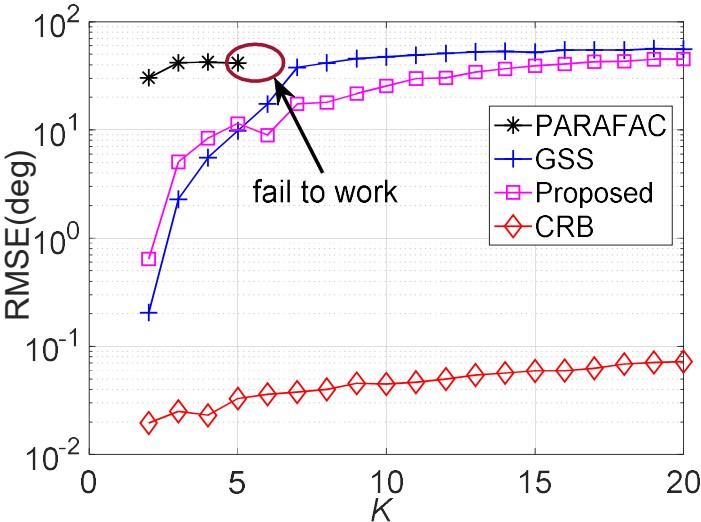

**Figure 7.** RMSE performance versus *K*.

## 6. Conclusions

In this paper, we have focused on the issue of coherent target positioning for ULA-configured EVS-MIMO radar. A spatial smoothing methodology has been presented, which de-correlates the signals via forward smoothing the data from the matched filters. Unlike the existing smoothing methods in the polarization domain, the proposed algorithm can provide 2D direction angles as well as polarization state of the targets. Numerical simulations show that the proposed algorithm can obtain automatically paired parameter estimation, and its estimation accuracy is proportional to SNR and the snapshot number.

**Author Contributions:** Conceptualization, writing—original writing, X.D.; methodology, Y.H. and C.L.; editing draft preparation, Q.W. All authors have read and agreed to the published version of the manuscript.

**Funding:** This work is supported by Shaanxi Provincial Key R & D Plan (No. 2020SF-166).

**Data Availability Statement:** Not applicable.

**Acknowledgments:** The authors would like to thank for the editors and reviewers for their efforts in improving this manuscript.

**Conflicts of Interest:** The authors declare no conflict of interest.

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
