# Peer review of "Coherent Targets Parameter Estimation for EVS-MIMO Radar"

_remotesensing, doi:10.3390/rs14174331_

Round 1

Reviewer 1 Report

The contribution claimed by the Authors of this manuscript reads:

a) “..  focuses on the coherent targets issue in electromagnetic vector sensor multiple-input multiple-output (EVS-MIMO) radar, and a spatial smoothing estimator is developed to estimate the multiple parameters.”.

b)  “It first recoveries the rank of the array data via forward spatial smoothing. Then, it estimates the elevation angles via the rotational invariance technique”

c) p.2. “After performing eigen-decomposition on the reduced covariance matrix, the estimation of signal parameters via rotational invariant techniques (ESPRIT) idea is adopted to estimate the elevation angles.”

d) “Combined with the vector cross-product method, the azimuth angles are obtained.”

e) p.8. “It has been stressed that our algorithm is able to deal with coherent target, thus the identifiability of the proposed algorithm should be analyzed from two aspects: the identifiability of coherent target and the identifiability of non-coherent target.”

The main weakness of this manuscript is the lack of clear identification of its novel contribution with respect to related publications and particularly [27]. Although the two simulated cases are adopted from [27], a comparison against the results of [27] is missing. This should include the strengths and drawbacks of the proposed algorithm against [27] in terms of identification accuracy, the behavior in the presence of noise and sensor position errors as well as computational time and required resources. The second weakness of this manuscript is the complete lack of relation to the involved physical phenomena and the related MIMO radar structure. Explicitly:

1) Instead of simply repeating the mathematical formulation [17], [18], [27] and the original references therein, the new contribution should be emphasized. The novelty claim should then be clearly supported by the advantages to be observed in the numerical results.

2) A physical reasoning is important for the Readers to understand the methodology and its capabilities. Thus, the quantities involved in the methodology and the numerical experiments should be related (i.e. interpreted one-to-one) to the physical structure of the MIMO radar.

3) It is important to explain how the data utilized are acquired, in the two cases of section 5. Are these the results of a simulation or measurements from a true physical structure? If as it seems are just simulation results, then what are the assumptions on which this model is based and what could be the deviation from true measurements?

4) Explicitly compare your results to those of [27] and clearly identify the possible adavantages and/or drawbacks.

5) In p.3 there is a claim as “ h stand for polarized phase difference”. What do you mean, is that just a phase difference between particular field components? How do you handle the possibly involved phase wrapping?

6) In p. 3 a statement claims that “A full EVS is composed of six collocated components: mutual orthogonal magnetic 99 loops (three) and electric dipoles (three), which sense the polarization state”. What do you mean? You assume an antenna - sensor capable of separately measuring all 6 components of the vector electric and vector magnetic fields? Meaning that this antenna – sensor will provide these 6 measured signals simultaneously? In my knowledge these types of sensors are very complicated, expensive and bulky and are only utilized in near field measurements. I’m very curious to see what the Authors have in mind.

            Overall, the novelty of this work is not clearly justified, especially with respect to [27], while it completely lacks any connection to the corresponding physical structures and the involved phenomena.

Reviewer 2 Report

The authors of the paper derived algorithm is developed for EVS-MIMO radar, which is able to deal with the coherent targets. Was performed eigen decomposition on the reduced covariance matrix, the ESPRIT idea is adopted to estimate the elevation angles. The VCP method was used to obtain the Tx/Rx azimuth angles. After the 2D-DOD and 2D-DOA estimation, the 2D-TPA and 2D- RPA were determined by using the methods least squares (LS) approach. The proposed algorithm was verified using computer tests. Proposed algorithm developed for EVS-MIMO radar I consider it correct. I appreciate the experiments performed using modeling and simulation. I consider the simulation conditions to be correct. The simulation results confirm that the algorithms created by the authors of the article they are functional. The topic of the paper is current. I recommend supplementing the conclusion of the paper with specific results obtained by simulation and comparing them with other methods of radar signal processing. I recommend publishing this paper after slight modifications.

Round 2

Reviewer 1 Report

In this revised version the Authors have acceptably addressed my previous comments, except no.2:

A physical reasoning is important for the Readers to understand the methodology and its capabilities. Thus, the quantities involved in the methodology and the numerical experiments should be related (i.e. interpreted one-to-one) to the physical structure of the MIMO radar.”

Addressing this concern the Authors elaborated only on its second part referring to the “numerical experiment”.

However, the comment was referring mainly to the methodology, asQ

A physical reasoning is important for the Readers to understand the methodology and its capabilities. Thus, the quantities involved in the methodology should be related (i.e. interpreted one-to-one) to the physical structure of the MIMO radar.”

Please, revise your manuscript accordingly.

Author Response

Sorry for the dealy. In this paper, the quantities involved in the methodlogy are listed as follows:

M: number of Tx EMVS

N: number of Rx EMVS

K: number of coherent target

M1: number of Tx EMVS in the subarray      

N1: number of Rx EMVS in the subarray

L: number of measurement

       In the last version, we have forgot to explain the physical means of M1 and  N1, in the final manuscript, we have added necessary descriptions.